
# Dynamically Coupling System Dynamics and SWAT+ Models using Tinamït: Applications of Modular Tools for Coupled Human-Water Systems Models

Joel Z. Harms[1], Julien J. Malard[1,2], Jan F. Adamowski[1], Ashutosh Sharma[3], Albert Nkwasa[4]

[1]Department of Bioresource Engineering, McGill University, Sainte-Anne-de-Bellevue, H9X 3V9, Canada
[2]Institut de recherche pour le développement (IRD), UMR G-EAU, Université de Montpellier, Montpellier, 34000, France
[3]Department of Hydrology, Indian Institute of Technology Roorkee, Uttarakhand, 247667, India
[4]Hydrology and Hydraulic Engineering Department, Vrije Universiteit Brussel (VUB), 1050 Brussel, Belgium

*Correspondence to*: Joel Z. Harms (joel.harms@mail.mcgill.ca)

**Abstract.** Participatory water resources management requires modeling techniques that are accurate and flexible, yet stakeholder-friendly. While different modeling frameworks offer advantages and disadvantages, System Dynamics (SD) models have seen sustained use as a stakeholder-friendly approach for water resources modelling. In contrast, physically-based models are more appropriate to model the hydrological components of coupled human-water systems. Proposed as a way to combine the relative strengths of both modelling paradigms, model coupling allows researchers to build participatory SD

models with stakeholders, while delegating the hydrological components of the overall model to an external hydrological model. Recently developed to facilitate model coupling, the Tinamït Python package presents an extensible outward-facing Application Programming Interface (API). It allows for the development of extensions (wrappers) that expand compatibility with different physically-based models. However, no watershed hydrological model has yet been connected to this API. In the present study, a socket and JavaScript Object Notation-based communication protocol was developed with the goal of

facilitating the coupling of models written in languages such as FORTRAN. This novel protocol served to develop a Tinamït-compatible wrapper for the hydrological model SWAT+, allowing it to be coupled to human-water SD models. The novel coupling protocol was then applied to a case study of Tanzania's Usa Basin. This approach provides the modeler with the benefits of both physically-based and SD models, thereby allowing the detection of potential far-reaching effects of policy decisions, within a system that remains flexible and easily adaptable to other watersheds.

## 1 Introduction

Given the worldwide threat to hydro-ecological systems from both natural and anthropogenic causes (Pahl-Wostl et al., 2013; Smith, 2003), accurate and holistic modeling of the long-term consequences of water resources management (WRM) decisions is a pressing need. The complexity of WRM problems in light of stakeholder issues mandates an approach that integrates physically-based components with socio- economic factors (Jakeman and Letcher, 2003). Integrated WRM models

are termed human-water models (Jeong and Adamowski, 2016). Considered an important and beneficial practice in current

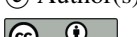



human-water modelling, Participatory Model Building (PMB) is often employed in creating such models (Butler and Adamowski, 2015; Halbe et al., 2014; Silgram et al., 2009; Zellner, 2008).

A stakeholder-friendly approach for water resources modelling, System Dynamics (SD) modeling is a commonly applied PMB tool. By visually representing aspects of system complexity that would otherwise be difficult to convey, this
approach can improve both stakeholders' and policy makers' accessibility of valuable information (Butler and Adamowski, 2015; Prodanovic and Simonovic, 2010). Having seen extensive use in integrated WRM (e.g., Fernández and Selma, 2004; Guo et al., 2001; Hassanzadeh et al., 2014; Jeong and Adamowski, 2016; Kelly (Letcher) et al., 2013; Qin et al., 2011; Saysel et al., 2002; Schlüter et al., 2019; Yeh et al., 2006), SD models were initially developed for the simulation of social systems in the context of industrial management. These models are often used to model complex, non-linear systems (Forrester, 1997;
Sušnik et al., 2012). Existing SD models for WRM involve the integration of hydrologic and socioeconomic processes (Langsdale et al., 2009; Sušnik et al., 2012); however, simulation of physical system processes is not easily achieved (Malard et al., 2017) and such models' representation of hydrologic processes remains a lengthy and site-specific process (Vamvakeridou-Lyroudia et al., 2008). Moreover, SD models lack the ability to easily incorporate spatial data and handle a system's spatial variability (Nikolic and Simonovic, 2015; Sušnik et al., 2012). Accordingly, hydrological components are
often simplified (Prodanovic and Simonovic, 2007). Nikolic and Simonovic (2015) therefore argued that a specialized hydrologic model is a necessity in integrated WRM modeling frameworks.

Specialized hydrological models provide hydrological variable values of interest to the WRM modeller with greater ease than a SD hydrologic model. This discrepancy in efficiency occurs because, given the necessary geological and hydrological variables, functions describing key hydrologic processes pre-imbedded in hydrological models can automatically
calculate the magnitude of such processes for any watershed. The Soil and Water Assessment Tool (SWAT) (Bieger et al., 2017; Nkwasa et al., 2020; R. Douglas-Mankin et al., 2010; Tuppad et al., 2011), a well-known, specialized, and widely applied hydrologic model, was developed to help water resources managers determine water quantity and quality in watersheds and river basins (Arnold et al., 1998). SWAT has been recently succeeded by SWAT+, which offers greater flexibility and overall capacities (Bieger et al., 2017). The capacities offered by hydrologic models like SWAT or SWAT+, and the suitability of SD
models for socio-economic modeling, suggest that an optimal WRM modeling framework could be achieved by coupling of these two model types.

Attempts to expand SWAT's capacities, through coupling with other models or applications, have been limited to improving its modeling of physical processes, but has yet to include the socioeconomic or participatory aspects of WRM (see: D. Betrie et al., 2011; Yang et al., 2011). Perhaps due to the briefness of its existence, there is yet no report of SWAT+ having
been coupled with other models. However, other hydrologic models have been coupled to SD models, e.g., the Upper Thames Systems Model (Gregersen et al., 2007). In these cases, coupling was achieved through translation of the constituent models into a single program. However, its specificity limited its application to a single watershed.

Coupling may also be achieved through the potentially more flexible file-based (Inam et al., 2017), scripting (Peck et al., 2014) or wrapper-based approaches (Gregersen et al., 2007). Malard et al., (2017) provide a review of current model





coupling approaches, and examples of standards for wrapping are provided by Salas et al., (2020). In the present study, a wrapper approach was employed since it is minimally invasive to the model itself (Gregersen et al., 2007), and was expected to offer maximum flexibility. However, wrapper approaches require a way of passing information between the two different models at runtime, a feat that becomes particularly complicated when the two models are written in different programming languages and must therefore run as separate processes on the computer. Inter-process communication in wrapping can be

achieved through a pipe-based approach (e.g., OpenMI; Gregersen et al., 2007); however, socket-based approaches are another option (Bulatewicz et al., 2013). The latter approaches are quite novel for this application and could provide potential benefits, e.g., shorter execution time (see Sect. 2.).

Most existing wrapping approaches have not been or cannot be used to couple to SD models. However, automatic SD model wrapping with external physically-based models may be achieved through the Tinamït API (Malard et al., 2017). In this

process, a physically-based model wrapper is developed only once for every physically-based model platform; thereafter, a few lines of code allow any models built upon the same platform to be coupled with any SD model. Tinamït has been previously applied to the evaluation of policy decisions for salinity management in Pakistan, using a coupled SAHYSMOD and SD model (Malard et al., 2017), as well as for agricultural management in predominantly Indigenous rural communities in Guatemala, coupling SD models and the Python Crop Simulation Environment (de Wit, 2022; Malard et al., 2020a, b). However, Tinamït

has not yet been applied to coupling an SD model with a hydrological model (e.g., SWAT+), thereby facilitating the integrated modelling of coupled human-hydrological systems. The concept of using Tinamït to couple SD and hydrological models has been presented, however, details of these earlier conceptualizations were mostly changed in the process of development (Harms et al., 2020, 2021). The present paper will present: (i) the development of a sockets and JavaScript Object Notation (JSON) based data-exchange Python package and its C/FORTRAN counterpart in SWAT+; (ii) the creation of a SWAT+ wrapper

using the functionality of the Tinamït API and the data-exchange Python package, to allow SWAT+ models to be automatically coupled to SD models; and (iii) the coupling and application of a SWAT+ and a human-water SD model, to holistically evaluate policy decisions and their potentially far-reaching effects in the Usa Basin, Tanzania, based on a SWAT+ model provided by Nkwasa et al., (2020).

## 2 Methodology

A Tinamït (2.0.3) compatible SWAT+ wrapper was created using the following steps:

1.   A python package (labelled 'tinamit-idm') employing socket functionality and including a generic JSON protocol, was developed to serve as the basis for data exchange between Tinamït wrappers and the SWAT+ model engine (Malard and Harms, 2022). (Python3)

2.   The SWAT+ engine was modified by adding C-language sockets allowing communication via the JSON-
protocol used by 'tinamit-idm' (Harms and Malard, 2022a). (Fortran, C)

3.   A SWAT+ wrapper was created using Tinamït 2.0.3 and tinamit-idm (Harms and Malard, 2022b). (Python3)





4.      The case-study model was coupled (Harms and Malard, 2022b). (Python3)

       A socket-based approach was chosen for the inter-process communication between the models instead of a pipe-and-
filter-based approach (*e.g.*, the OpenMI standard) (Gregersen et al., 2007; Bulatewicz et al., 2013). The socket-based approach
       was expected to save execution time, since no files are written to disk. A further advantage of our approach was that the JSON-
       protocol used here virtually eliminated the need for language inter-operability. Almost any programming language (see *JSON*
       (n.d.)) can simply create and parse JSON messages (ECMA, 2017). A conceptual overview of a complete coupled model and
       its parts is presented in Figure 1.

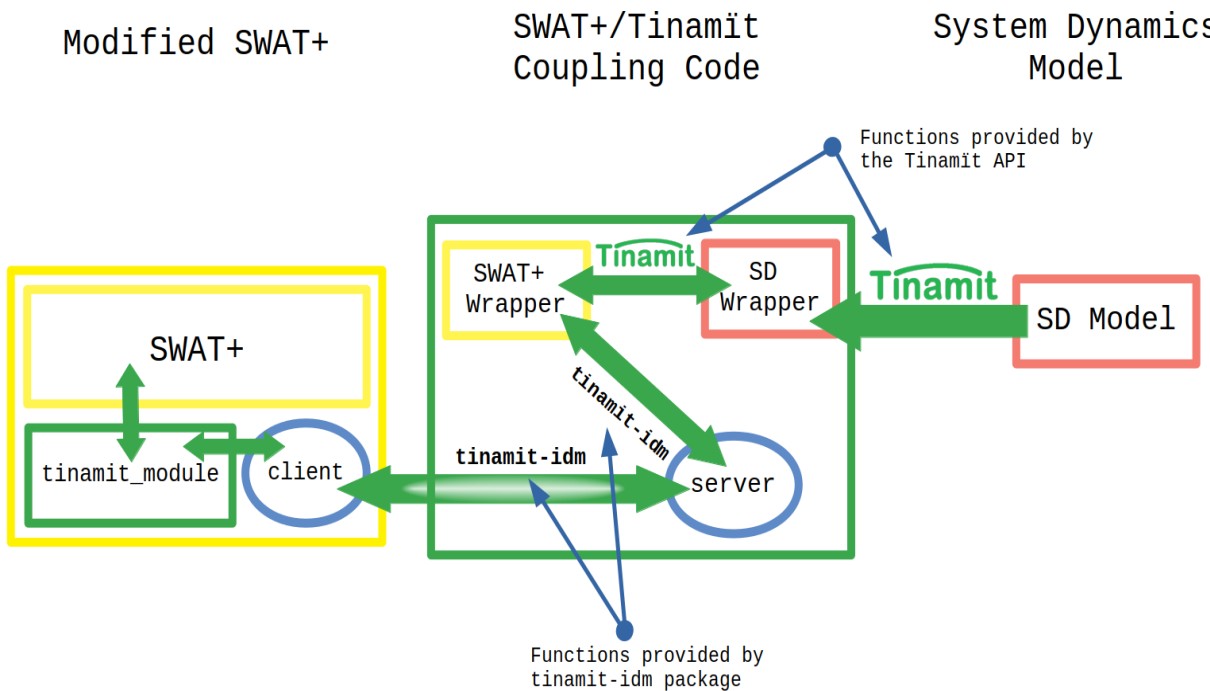

**Figure 1: A schematic summary of the coupling process using the pre-existing Tinamït API and the newly developed tinamit-idm Python package. This schematic is adapted and modified from Harms et al., (2020, 2021).**





## 2.1 tinamit-idm

A generic socket-based python package was developed to provide basic functions that any models interchanging
information during run-time might require (Malard and Harms, 2022). The functions 'cambiar', 'recibir', 'incrementar',
'finalizar' and 'cerrar' define how to provide data, receive data, increment the model by a certain number of time-steps, finalize
the model and close the model, respectively. The JSON protocol was of critical importance for these functions. Five distinct
types of JSON message types were defined, corresponding mostly to the key functions provided by tinamit-idm. The definition
of this JSON-protocol was kept as general as possible to give it the necessary flexibility to be implemented as a standard for
all socket-based inter-model communication using tinamit-idm and Tinamït. The JSON messages are detailed in Table 1.

**Table 2: Presentation of the generic JSON protocol. The context and order of these messages is more clearly illustrated in Figure 1.**

| Message Type | JSON key | Values | Description |
|---|---|---|---|
| **Mensaje Cambiar** | *This message, sent by the 'cambiar' function, tells the model to receive some variable's current values, and is followed by the variable value.* | | |
| | "tipo" | "cambiar" | Specifies the type of command |
| | "tamaño" | [integer] | Holds the length of the array buffer of the variable value |
| | "var" | [name strings] | Holds the name string of the variable to be received |
| | "tipo_cont" | [data type string] | Holds string of data type of the variable values |
| | "forma" | [array shape string] | Holds shape string of the array of the variable values |
| | "precisión" | [integer] | Defines the number of decimal places that non-integer numerals may be rounded to. |
| **Mensaje Recibir** | *This message is sent by the 'recibir' function, it asks the model to send the current value of a variable, the model responds with a 'Mensaje Resultado'.* | | |
| | "tipo" | "leer" | Specifies the type of command |
| | "var" | [name strings] | Holds the name of the variable that the model should send |
| | "precisión" | [integer] | Defines the number of decimal places that non-integer numerals may be rounded to. |
| **Mensaje Resultado** | *This is the only message sent by the model itself, it is only sent in response* | | |





| | | | |
|---|---|---|---|
| | | | to a 'Mensaje Recibir' and is followed by the value of the expected variable. |
| | "tamaño" | [integer] | Holds the length of the array buffer of the variable value |
| | "tipo_cont" | [data type string] | Holds string of data type of the variable values |
| | "forma" | [array shape string] | Holds shape string of the array of the variable values |
| **Mensaje Incrementar** | *Sent by both the 'finalizar' and 'incrementar' functions: 'incrementar' asks the model to run for a specific amount of time or number of steps of its main loop, whereas 'finalizar' asks the model to run to completion and not receive any further messages.* | | |
| | "tipo" | "incr" | Specifies the type of command |
| | "n_pasos" | [integer] | Number of time-steps the model should take before reporting back; in the case of 'finalizar' this number is 0 and indicates that the model should run to completion |
| **Mensaje Cerrar** | *This message is sent by the 'cerrar' function. It signals to the model that the simulation is done and closes the connection.* | | |
| | "tipo" | "cerrar" | Specifies the type of command |

## 2.2 Modification of SWAT+

To give SWAT+ the ability to exchange data via sockets, SWAT+ was modified as follows (Harms and Malard, 2022a):

Argument handling was initially added so that one might specify whether SWAT+ was to be executed alone, or connected to Tinamït. Arguments can be added to the command-line call of the swatplus.exe file; the execution file can be called "swatplus [port] [host]" (where port specifies the port to which SWAT+ should connect and host specifies the host IP, usually 127.0.0.1). One may still call "swatplus" to run only the SWAT+ model by itself.

Secondly, a FORTRAN module implementing the methods required for communication with the tinamit-idm package was added. This module was named the "tinamit_module" and is called mainly from the "time_control" module that manages SWAT+'s main simulation loop. The "tinamit_module" gives SWAT+ the ability to pause, receive messages from and send messages to other models coupled via Tinamït. Furthermore, a mechanism is implemented here that allows SWAT+ to run as





many time-steps as the coupling code specifies before re-communicating with the Tinamït-wrappers. This alleviates the need to unnecessarily pause the simulation of SWAT+ after every time-step to communicate with Tinamït, and allows for a more efficient execution of the simulation depending on the data-exchange interval chosen by the modeler. It also allows Tinamït to balance the varying time-step lengths of the different models. The "tinamit_module" makes use of a C-file ("socket.c"), which provides it with socket creation as well as reading and sending functions (hereafter referred to as the C-client). The sub-

processes and functions defined in the "tinamit_module" and the C-client as well as the workflow between them are illustrated in Figure 2.

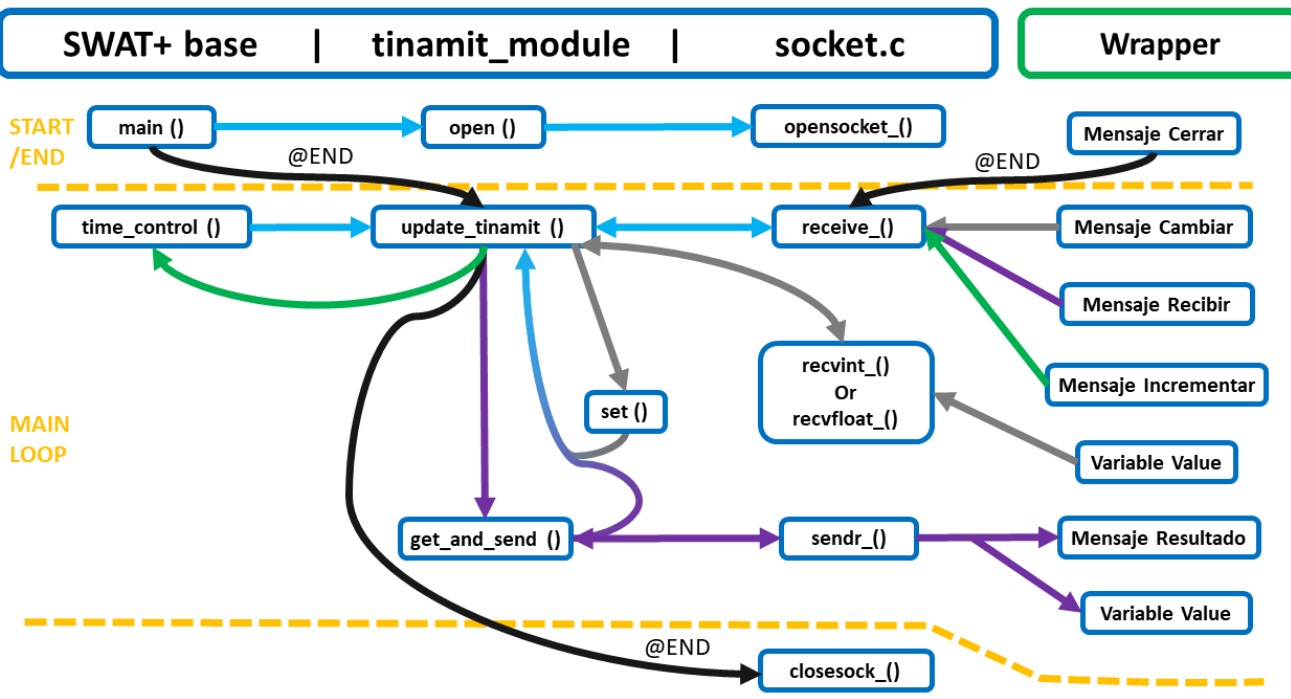

**Figure 2: Graphic representation of the expected SWAT+ model behavior and communication with the wrapper during coupled model execution. The 'set' loop is shown in grey, the 'get_and_send' loop is shown in purple, the 'incrementation' pathway which**

**informs the model engine to run for a specified number of time-steps is shown in green and the closing sequence is in black. All other pathways of information and function calls, including those used for multiple above-mentioned pathways, are shown in cyan.**

SWAT+ model execution and socket client are initiated by the main routine. Later, 'update_tinamit' is called from within the 'time_control' subroutine at the first time-step and thereafter at every specified interval. One 'get-message' or

'Mensaje Cambiar' is sent per connected variable (by the wrapper) so the 'get_and_send' loop will be repeated until all requested current variable values have been sent to the wrapper. Then, after the connected model (in this case the SD model) is run for a time-step, the 'set' loop will be performed until updated variable values have been received (for those variables





that are specified as inputs to the SWAT+ wrapper). Thereafter, the 'Mensaje Incrementar' or 'advance-message' is sent to continue the model execution for a specified number of time-steps and the process repeats until SWAT+'s 'time_control'

function is completed. Finally, 'update_tinamit' is called by the main routine as part of the closing sequence at which time the 'Mensaje_Cerrar' or 'closing-message' will be sent to close the socket and end the run. Only standard, anticipated, or recommended behaviour is shown, as it is defined within the wrapper (see Sect. 2.3). However, the sequence of messages could be altered, such as closing the client during the run. This may be specified by users according to their needs should they decide to create their own wrapper.

**2.3 Creation of the SWAT+ Wrapper**

A SWAT+ wrapper was created using the Tinamït and the tinamit-idm packages (Harms and Malard, 2022b). This wrapper is openly available on GitHub, as are the other source codes for the work mentioned in this paper. Since SWAT+ is a physically-based model, a subclass of the Tinamït wrapper class "ModeloBF" class was created, called "ModeloSWATPlus". The following methods were defined for the "ModeloSWATPlus" subclass:

1.  __init__(): this method creates an instance of the "ModeloSWATPlus" and initializes the socket connection as well as the model variables.

2.  unidad_tiempo(): returns the unit of time of the model for SWAT+. This was set to daily.

3.  iniciar_modelo(): defines how the wrapped SWAT+ model will be called to start running and sets up a working directory for the model and ensures that the other requirements for data exchange are met.

4.  incrementar(): defines how the model will be advanced during its run and how    data exchange with the model may occur after a simulation step.

5.  paralelizable(): defines whether SWAT+ is parallelizable in different Python threads, which it is, if run using the wrapper.

6.  cerrar(): ends the model run and stops the model engine. In this case it is also responsible for ending the connection between the model and the Tinamït wrapper.

7.  instalado(): checks whether a SWAT+ .exe or engine is installed and specified.

Steps 1-7 must be implemented by any physically-based model wrapper created in Tinamït (Malard et al., 2017). The following three are convenience functions that the authors deemed useful for any SWAT+ model in general and more

specifically for case-study model execution:

1.  imprimir_usos_de_tierra(): prints the land use types defined for the current project so that the user does not necessarily need to read the SWAT+ input files to know which land uses are identified by which numerical value.





    2.  deter_área_de_hrus(): reads SWAT+ input file "hru.con" to determine and store the size of all the specified Hydrological Response Units (HRUs), which should not change during simulation and can therefore be cached.

    3.  agrupar_usos_del_suelo(): allows the user to group land uses into specific land use types for simpler simulation of land use changes.

    This wrapper is currently an independent project but will be integrated alongside tinamit-idm into the next major Tinamït version and will then be even easier to use. At the time of the writing, Tinamït 3.0.0 is still in development.

**2.4 Case Study**

    A case study was built using a SWAT+ model validated for Tanzania's Usa Basin (Nkwasa et al., 2020), and a SD
model developed by the authors. Details on the SWAT+ model setup and validation can be found in (Nkwasa et al., 2020). To summarize, the SWAT+ model set up for Usa basin (240 km$^2$) mainly utilized local data with a specific focus on agricultural practices in the basin. The model validation included Leaf Area Index (LAI) and water balance components (evapotranspiration, surface runoff and groundwater). The case study SD model was built to show potential far-reaching effects of a hypothetical nitrogen-fertilizer subsidy on total agricultural land use and water quality in the basin. It is important to note
that the case study and model parametrisation were chosen as a generic example of the sociological and hydrological insights made possible by coupling SWAT+ with an SD model through our approach, which is the main scope of the paper. Extensive and participatory model development and calibration with stakeholders from the Usa Basin would be needed to use the coupled model in actual policy design. As such, the results of the case study as presented in this paper should be understood as an example of potential unexpected policy impacts that may be detected by using SWAT+ and SD model coupling, and not as a
prescriptive policy recommendation for the Usa Basin.

    Vensim (Eberlein and Peterson, 1992) was used to develop the SD portion of the model, as well as the causal loop diagram of the overall coupled model. A diagram of the overall model (Figure 3) shows it to consist of four main loops. Given the coupling, it is difficult to call these reinforcing or balancing loops since they could shift behaviours during model execution. Therefore, "r" and "b" were used to note which loops seemed likely to display reinforcing or balancing behaviours,
respectively. The "Banana Production" (rBP) and "Corn Production" (rCP) loops were assumed to generally be reinforcing, since an increase in agricultural land would likely also lead to an increase in production of banana and corn crops. The "Cultivation Area" loops ("Corn Cultivation Area" loop: bCCA, "Banana Cultivation Area" loop: bBCA) acted as balancing loops since an increase in agricultural land would lead to a greater need for inputs and therefore increase production cost. The strength of the balancing loops was modified by the hypothetical fertilizer subsidy, since the production cost per area would
decrease, thereby lowering the effect of cultivation area on net income.

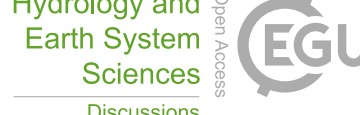

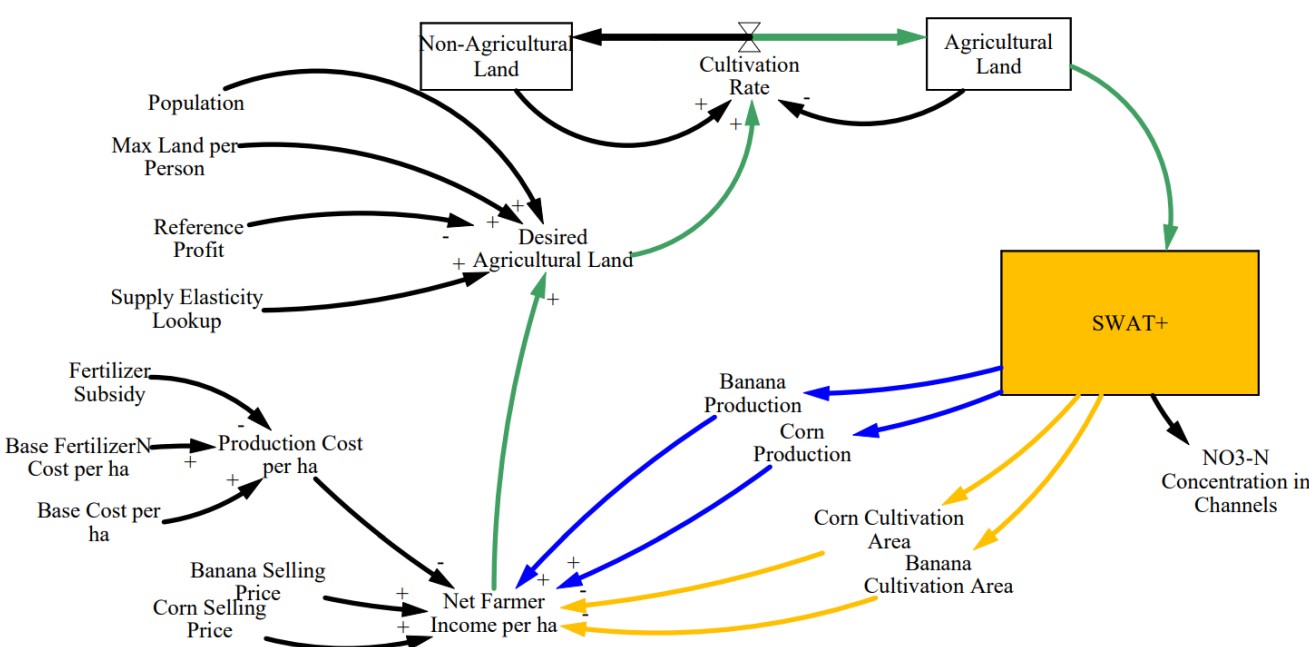

**Figure 3: Representation of the case-study model as a causal loop diagram. The main loops are represented as: blue for the loops of "Banana Production" (rBP) and "Corn Production" (rCP), yellow for the loops of "Corn Cultivation Area" (bCCA) and "Banana Cultivation Area" (bBCA), green for the overlapping parts of all loops. This figure is adapted and substantially modified from Harms et al., (2021).**

The SD model described here represents a very simplified view of how a policy, such as a hypothetical fertilizer subsidy, might affect water quality in the basin. The fictional 'Fertilizer Subsidy' might be implemented to maintain agricultural activity in the Usa Basin or to increase farmer income. The implementation of such a subsidy would lower the 'Production Cost' by covering a portion of the input costs, which would, in turn, increase the 'Net Farmer Income'. The greater 'Net Farmer Income' would lead to an increase in the 'Desired Agricultural Land', which, in turn, would increase the 'Cultivation Rate', and turn more land into 'Agricultural Land'. The value of 'Agricultural Land' was then passed on to the SWAT+ model. The amount of agricultural land given by the SD model was compared to the total area of HRUs in the SWAT+ model used for agricultural land use types. At every time-step, the wrapper made a calculation to determine whether it was necessary to switch the land use of any HRU of the SWAT+ model to better approximate the current extents of agricultural and non-agricultural lands. Although the corn and banana crops were the main contributors to the total agricultural land considered in the SD model, the Usa Basin SWAT+ model in fact contained multiple different types of agricultural and non-agricultural land-uses. To simplify the SD model, the land use types were assigned to two groups, namely agricultural or non-agricultural, using the 'agrupar_usos_del_suelo' function mentioned in Sect. 2.3. When the land use of an HRU was to be changed, a land use was selected from the appropriate group and assigned to this HRU. The selection happened





programmatically, favouring land uses with a high popularity in each group (See Nkwasa et al., (2020) for a complete list of land uses in the Usa SWAT+ model).

SWAT+ completed the main reinforcing loops (rBP, rCP) by returning the total basin-wide 'Banana Production' and 'Corn Production', which were expected to increase with increasing area dedicated to agricultural land. Values for 'Banana Cultivation Area' and 'Corn Cultivation Area' were also returned by the SWAT+ wrapper portion of the model. These were also expected to increase with an increasing expanse of agricultural lands. The inclusion of SWAT+ was of further importance since SWAT+ provided the 'NO3-N Concentration in Channels', an indicator of water quality in this case study. This allowed an analysis of the risk to water quality degradation posed by the policies tested, i.e., 0% fertiliser subsidy (Base-line scenario), 45% subsidy (Subsidy scenario) and 90% subsidy (High-subsidy scenario). Three simulations were performed for each scenario.

We restate here that the values of the non-SWAT+ variables were chosen to give an interesting result and illustrate processes that might occur in real-life situations. Many simplifying assumptions were applied, most importantly in only considering the production of corn and bananas as well as assuming a constant fertilizer cost per ha of cultivated land.

The source code, as well as the SD model used, are provided (see Sect. 6.).

## 3 Results and Discussion

The results of the coupled simulations are given in Figure 4 and include the mean of three runs along with the standard deviation. Variation arose between runs due to the land-use change algorithm used in the wrapper. The land-use type assigned to an HRU where land-use was changed, was chosen at random from a group of appropriate land-uses (either non-agricultural or agricultural in this case) with a weighted probability to favor land-uses already dominant in the watershed.

Increasing the subsidized fraction of nitrogen fertilizer achieved the goal of maintaining agricultural production in the area (Figure 4). It not only stabilized but reversed the decreasing trend encountered in the baseline (0% subsidy) scenario and increased the amount of agricultural land under the 45% and 90% subsidy scenarios. However, it also increased the nitrate-nitrogen concentration in the channels of the basin. Through the coupling of the SWAT+ and SD variables, this relationship could be easily explored. Similarly, the water resources manager may obtain any hydrological value that SWAT+ calculates (such as river flow and nitrate nitrogen content, as used in this study), without developing any new algorithms for these processes. This allows the researcher to focus on the development of the model to represent the socioeconomic relations in a participatory manner.





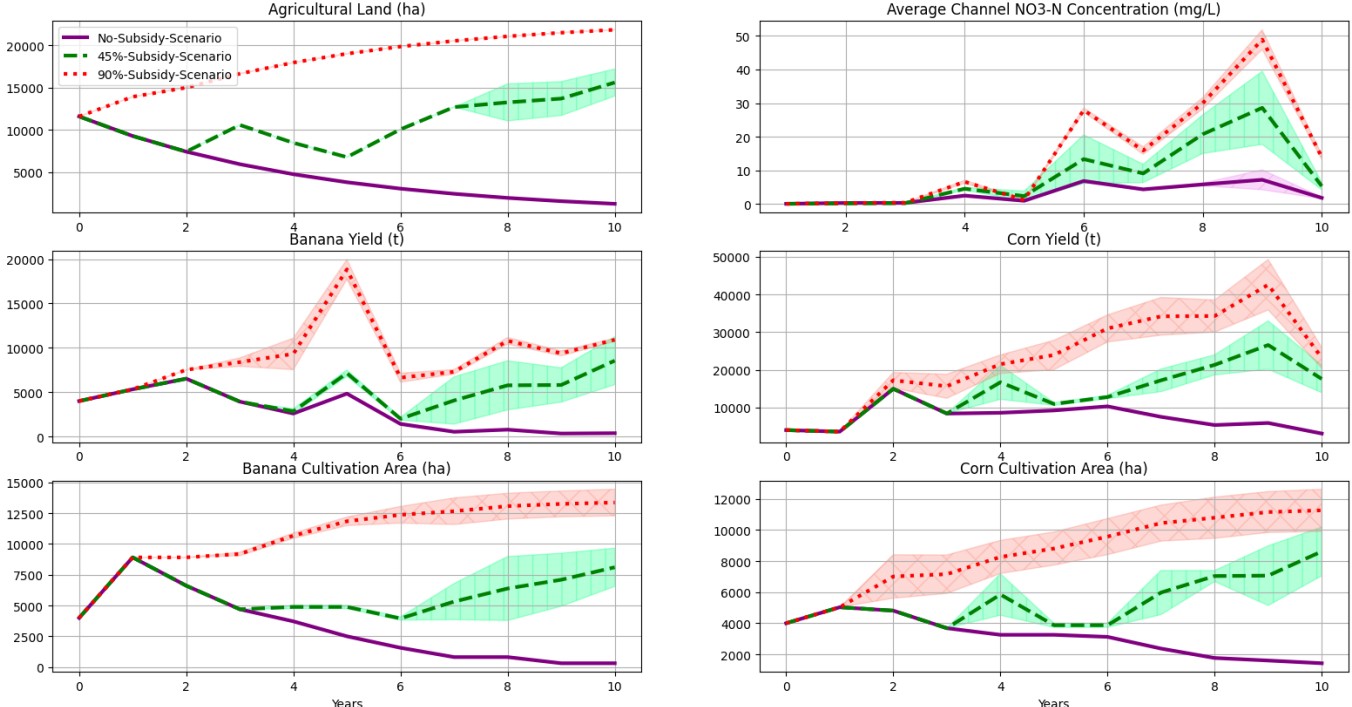

**Figure 4: The results from the coupled models with 0%, 45% and 90% subsidized fertilizer. Three runs were performed; mean and standard deviation (shaded area) are plotted.**

The model coupling approach shown here empowers researchers and policy makers by giving them access to both an inclusive understanding of the system through participatory model building, and the physical consequences of possible paths of action. More than that it empowers stakeholders by allowing more focus on their involvement in decision-making. Furthermore, this approach works for any watershed where a SWAT+ model can be created, and similarly any SD model. Although the initial modification of SWAT+, the development of the server and client parts for inter-process communication, and the construction of a SWAT+ wrapper were tedious and lengthy endeavors, the ability to couple any SD models to any other SWAT+ model far outweighed the time and work required. This was demonstrated in earlier work with Tinamït (Malard et al., 2017, 2020a, b). Through this method, the researchers were able to focus on representing the socio-economic processes in the watershed, or in potentially involving further stakeholders without sacrificing the hydrologic accuracy of the overall model.

Additionally, since a generalized JSON protocol was used in concert with socket technology, inter-process communication was achieved between diverse languages (Python, Fortran, and C in this case), without the need for writing and reading files to disk. Other coupling methods which have employed file and pipe-based approaches (Gregersen et al., 2007; Inam et al., 2017) or translated the entire model into a single programming language (Prodanovic and Simonovic, 2010),





have shown potential shortcomings, e.g., longer execution time, extensive implementation work, and a resultant lack of flexibility. Since no files were written or read during execution when the socket-based data exchange was in use, model execution time was likely to benefit. This technology could furthermore be adapted to a multitude of other FORTRAN (or C) based models without major modifications to the client part of the socket code (i.e., the 'socket.c' file), as the C socket client is as generic as the JSON protocol used by tinamit-idm. The server code in tinamit-idm, especially in conjunction with Tinamït, lays the groundwork for developing wrappers for models written in any language that can interpret JSON, and will simplify the development of other physically-based model wrappers.

## 4 Limitations

Compared to OpenMI, Tinamït has neither the same status nor history of use and trust and does not yet support a similar range of models. Therefore, the OpenMI standard may currently have an advantage when different physically-based models are to be coupled for a project. However, Tinamït is unique in its ability to create wrappers for SD models, which are some of the most popular tools for integrated WRM models or, more broadly, participatory model building. Additionally, the work presented here is likely to improve the usefulness of Tinamït, not only through the possibility of easily integrating SWAT+ into a participatory WRM model, but also because it provides standards, methodologies and examples that simplify the addition of further models to Tinamït.

Currently, the SWAT+ coupling-capable version is only available for Linux systems, although Windows and MacOS versions are in development. During development, the Windows version of the socket code in the modified SWAT+ version created difficulties during compilation. These difficulties were exacerbated by differences between Windows-sockets and C-sockets. The MacOS version is nearing the end of its development but at the time of writing, the Linux version of the modified SWAT+ was the only functional version available. However, all Python packages presented (i.e., Tinamït and tinamit-idm) are functional on all current major operating systems.

A selection of about 190 numerical type SWAT+ variables have been made available for coupling, including some utility functions and variables specific to the case study; however, there are likely functions and variables that other modelers might find useful to couple that have not been included. Therefore, we expect and encourage modelers to not only use, but collaboratively modify and improve the current codes and expand the number of supported variables and functionalities to their needs, as we continue to do the same. This is possible since all code mentioned in this project is published in open-source repositories (see Sect. 6.).

## 5 Conclusion

To make holistic decisions for human-water systems, both the socio-economic and the physical systems need to be understood. Since different models and modelling platforms have different strengths, combining multiple models through

coupling may provide a more complete modeling platform. Coupling processes can be difficult and coding intensive, so this study developed a simple approach specialized for participatory models based on Tinamït. The work presented in the present paper provides algorithms and functionalities, the tinamit-idm package and a modified SWAT+ executable with socket functionality. These can function as templates for the development of future coupled Fortran models. The Fortran compatible C-client from the modified SWAT+ executable can potentially be applied to make any Fortran model much more easily couplable with Tinamït than it would be in this initial project. The use of sockets and the JSON-protocol is a promising step as it largely bypasses the need for inter-language-operability and potentially improves execution-time compared to other model coupling platforms. Expansion of the abilities of Tinamït makes it more useful for participatory WRM as well as other participatory modeling work. In the case study presented here, the coupled model was able to portray the overall results of both the human, socio-economic, and the physical resource systems at play, with considerable ease. The modeler was able to obtain results for hydrologic variables without having to define all the relationships manually in the SD model, thereby allowing a focus on representing socio-economic processes.  In closing, we hope that the code and technology presented in this paper will contribute to improving participatory WRM modelling work, and make it easier to apply to real-world decision-making scenarios while also providing developers with ideas and templates for the further development of Tinamït and other model coupling platforms.

## 6 Code Availability

The source-codes of the software presented in this paper including the case-study are available, current archived versions are cited in the text (Harms and Malard, 2022b, a; Malard and Harms, 2022), and any updated versions are available publicly on GitHub.

## 7 Author Contribution

Conceptualization for this work was done by JM, JA, AS, the methodology was designed by JM and JA. Funding was acquired by JH, JM, JA. JA administered the project while supervision was carried out by JM and JA. Software was developed by JH and JM. AN provided modelling resources for the development of the case-study. Code-testing and validation was carried out by JH who was also responsible for the visualization of the work and the creation of the figures. The original draft was prepared by JH, JM and JA while all authors were engaged in review and editing of the final draft.

## 8 Competing Interests

The authors declare that they have no conflict of interest.



**9 Acknowledgements**

This research was supported in part by a National Sciences and Engineering Research Council of Canada (NSERC)
Undergraduate Student Research Award (USRA), supplemented by the Fonds de recherche du Québec (FRQNT), held by Joel
Harms. This research was further supported in part by a grant from NSERC held by Jan Adamowski. Ashutosh Sharma was
supported by the Shastri Indo-Canadian Institute (SICI) under the Shastri Research Student Fellowship (SRSF) scheme to
carry out this research at McGill University.

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
