# Peer review of "Dynamically Coupling System Dynamics and SWAT+ Models using Tinamït: Applications of Modular Tools for Coupled Human-Water Systems Models"

_Hydrology and Earth System Sciences, 2022_

## Author Response (AR1)

**Reviewer 1.**

Dear Reviewer,

Thank you for your comments! We will address them in detail below:

The present work details a methodology to couple a hydrological model/software (SWAT+) with a system dynamic (SD) model/software that rend the human-water interaction. The overall goal is to provide a piece of machinery in which the hydrological component of the human-water interaction is represented with a higher degree of fidelity. The paper is well written and the diverse aspects of the coupling, as well as, the adopted choices are well explained and justified. Limitations are also well recognized.

Thank you for a positive evaluation of our study and your time to provide critical comments to improve the manuscript!

I do personally see the present work as a technical note, rather than a research paper, given the high degree of technicality and focus on the practical aspects of the developed methodology/integration of software.

Similarly to Khan et al. (2017) and Zhang et al. (2017), both published in HESS as research papers, we introduce a novel coupling framework. Our paper, as does theirs, includes a case study to show the utility of the newly developed methodology that highlights some possible far-reaching consequences of policy decisions. Furthermore, as our paper goes into details of general and more specific aspects of model coupling, has an extensive literature review of System Dynamics based socio-hydrological models as well as a case-study, we believe that the manuscript could be considered for publication as a research article. However, since both reviewer 1 and 2 suggest publishing our paper as a technical note, we won't object to changing to a technical note if the editor prefers this option as well.

**References:**

Khan, H. F., Yang, Y. C. E., Xie, H., & Ringler, C. (2017). A coupled modeling framework for sustainable watershed management in transboundary river basins. *Hydrology and Earth System Sciences*, *21*(12), 6275–6288. https://doi.org/10.5194/hess-21-6275-2017

Zhang, L., Lu, J., Chen, X., Liang, D., Fu, X., Sauvage, S., & Sanchez Perez, J.-M. (2017). Stream flow simulation and verification in ungauged zones by coupling hydrological and hydrodynamic models: A case study of the Poyang Lake ungauged zone. *Hydrology and Earth System Sciences*, *21*(11), 5847–5861. https://doi.org/10.5194/hess-21-5847-2017

Minor comments

Lines 27-28: 'The complexity of WRM problems in light of stakeholder issues mandates an approach that integrates physically-based components with socio- economic factors' the sentence is a little bit involved, please consider to reformulate it.

Thank you for pointing this out. We will reformulate it to for the revised version: "To capture the complexity of WRM stakeholder issues in models, approaches that integrate physical and socio-economic components are necessary."

Lines 47-48: 'Specialized hydrological models provide hydrological variable values of interest to the WRM modeller with greater ease than a SD hydrologic model.' With great ease (see also Lines 266-268) or great degree of fidelity with the 'true' hydrological variables?

This sentence refers to the ease with which a certain hydrological value may be ascertained from each model type. For hydrological models all formulas are pre-built and all variable values from a simulation are accessible in the SD. As we describe in lines 48-50: "This discrepancy in efficiency occurs because, given the necessary geological and hydrological variables, functions describing key hydrologic processes pre-imbedded in hydrological models can automatically calculate the magnitude of such processes for any watershed." For SD hydrologic models, a new SD model, or sub-model, needs to be built from scratch for a specific basin and only variables that were explicitly built into this model may be accessed, such that any additional variable of interest requires modification of the model structure. This is described in more detail also in lines 40-45: "Existing SD models for WRM involve the integration of hydrologic and socioeconomic processes (Langsdale et al., 2009; Sušnik et al., 2012); however, simulation of physical system processes is not easily achieved (Malard et al., 2017) and such models' representation of hydrologic processes remains a lengthy and site-specific process (Vamvakeridou-Lyroudia et al., 2008). Moreover, SD models lack the ability to easily incorporate spatial data and handle a system's spatial variability (Nikolic and Simonovic, 2015; Sušnik et al., 2012). Accordingly, hydrological components are often simplified (Prodanovic and Simonovic, 2007)." .

We are not referring to fidelity or accuracy of each model type since either model type may have different accuracy levels depending on the case study and situation.

Lines 57-58: 'Attempts to expand SWAT's capacities, through coupling with other models or applications, have been limited to improving its modeling of physical processes' not very clear which aspect of the modeling: fidelity? Robustness? A bit too general the word modeling here.

What we meant here is the improvement of SWAT's ability to model more physical processes than it was originally built for or adding more detailed components to increase fidelity and robustness. We will adjust the sentence to clarify that we mean scope, fidelity and robustness in the revised version. The new sentence will be as follows: "Attempts to expand SWAT's capacities, through coupling with other models or applications, have been limited to expanding it's scope, fidelity or robustness in modelling various physical processes"

Line 120: 'To give SWAT+ the ability to exchange data via sockets, SWAT+ was modified as 120 follows (Harms and Malard, 2022a):' better close the sentence with a dot or modify the subsequent paragraph as a part of a list.

We will change the colon to a period for the revised version.

Lines 246-247: 'The results of the coupled simulations are given in Figure 4 and include the mean of three runs along with the standard deviation.' I am aware that this is not the purpose of the paper, but I would say that 3 trials are rather insufficient to robustly assess an expected value and even more so a standard deviation.

We have increased the runs to 10 now to add robustness, and will adjust the figure and the text for the revised version. However, we still want to emphasize that this case-study was not explored with the intention of others using it to make policy decisions for the Usa basin. We

state this in the text also on lines 198-200: "the results of the case study as presented in this paper should be understood as an example of potential unexpected policy impacts that may be detected by using SWAT+ and SD model coupling, and not as a prescriptive policy recommendation for the Usa Basin".

Lines 266-268: 'Although the initial modification of SWAT+, the development of the server and client parts for inter-process communication, and the construction of a SWAT+ wrapper were tedious and lengthy endeavors, the ability to couple any SD models to any other SWAT+ model far outweighed the time and work required.' I am happy it was worth it.

Thank you! We appreciate your feedback.

**Reviewer 2.**

Dear Reviewer,

Thank you for your comments, we appreciate your feedback. We will address all comments in detail below:

General comments:

The coupling of social models with physical models is an emerging field of research and software development for supporting human-environment interaction. The article provides a valuable and interesting contribution as the two models System Dynamics and SWAT+ are widely used in their fields. SWAT+ is a newer version of the well known SWAT model, which is a hydrological model often used in watershed management context. In that sense the article is of importance for the scientific community, and it adresses recent topics of interest across disciplines. The chosen wrapper approach for coupling SD and SWAT+ is reasonable and justified.

The manuscript is overall well written and understandable. The objectives (i) and (ii) of the paper are of purely technical nature and (iii) serves for proving the functionality with a case study. As the model coupling is an original contribution, a case study is very useful, but does not make the article being a research study. The more suitable manuscript type is "technical note", and it will be reviewed as such in the following.

The level of innovation is seen as sufficient for a technical note within the journal, as the topic is seen as adequate.

Regarding the "Technical Note" comment we want to point to our response of a similar comment made by Reviewer 1. We believe that the article could be published as a Research Article, however, if the editor considers the publishing as a "Technical Note" more appropriate we will not object to publish the paper as a "Technical Note".

Specific comments

L 55: "optimal" is a very optimistic if not exclusive statement - there might be other useful approaches of coupling social models with physical models. The authors did not make a comprehensive literature survey of other methods in coupling physical models with social models (e.g., agent based modelling, theory of planned behaviour, several varieties of DSS, …). They focus on SD and SWAT+, which is acceptable for a technical note about their solution for modelling human-environment interaction in WRM.

We will change the wording to not use such strong words like "optimal" but instead we will say "suggest that an improved WRM modeling framework could be achieved by coupling of these two model types".

General remark regarding the code: using another language than English, even using special characters in a code is not very convenient for the international community. The authors may consider replacing Spanish terms with English ones for easier adoption by international researchers.

This is done for better interoperability with Tinamit, which is in Spanish. As we mention in the paper the next version of Tinamit is expected to include the present wrapper and tinamit-idm module which would make Tinamit less usable if we had a mix of languages. We have included Spanish and English comments in the wrapper code to make it more widely usable. Furthermore, the coupleable SWAT+ is English and only the parts that contact Tinamit are in Spanish to keep the interoperability.

Figure 3: the diagram shows the role of SWAT as a crop model, where the yields are finally linked with socio-economic figures and decisions of farmers. A linkage to hydrology is not clearly visible. The leaching of N from the model area is leaving the coupled system without serious interaction in defining environmental policies, or including environmental criteria into the decisions of farmers. As there is a fertilizer subsidy defined, the linkage between that subsidy and it's effect (by changing N leaching) should be made more clear.

We will make the link between the subsidy and N-leaching more clear in the revised version. The coupling of SWAT+ presented here is useful to obtain variables, that are readily available in SWAT+, in a SD model. This includes both crop-related and hydrologically related variables to simplify human-water-systems modelling. The case-study manages to show this expansion in the breadth of the modelled variables, even though not all variables are, in this simple case-study, included in the feedback loops. Additionally, SWAT+ itself is using meteorological and hydrological variables to calculate the crop yield. To address this line 265-281 will be as follows: "By decreasing production costs, the subsidies encourage increased investment in agriculture, the conversion of more land, and so eventually increased leaching through a socio-economic pathway. This happens because the policy tilts the economic equilibrium point of optimal agricultural land towards more agricultural activity; the SWAT model then links this dynamic to environmental pollution through the increasing nitrate-nitrogen concentration in the channels of the basin. This relationship was easily explored through the coupling of the SWAT+ and SD variables. Generally, the water resources manager may obtain any hydrological value that SWAT+ calculates (such as river flow and nitrate nitrogen content, as used in this study), without developing any new algorithms for these processes. In this case-study hydrologic variables are incorporated directly, through the Nitrogen concentration variable, and indirectly, as SWAT+ uses hydrological variables internally to model the crop yield."

Code availability: the authors should include a link to the Github or Zenodo page where the code will be finally published.

We will include a link to the GitHub page as you suggest; we already cite the Zenodo page which has the archived code. The GitHub page will be what is updated and the Zenodo page is archived for reference.

Technical corrections

L 68: "feat" means "feature"?

No, feat as in an accomplishment, we will reword this to "accomplishment" for clarity.

L 116: table 2 should be labelled table 1

Thank you for pointing this out, we will re-label that.